# TurtleBench: Evaluating Top Language Models via Real-World Yes/No Puzzles

## Abstract

As the application of Large Language Models (LLMs) expands, the demand for reliable evaluations increases. Existing LLM evaluation benchmarks primarily rely on static datasets, making it challenging to assess model performance in dynamic interactions with users. Moreover, these benchmarks often depend on specific background knowledge, complicating the measurement of a model's logical reasoning capabilities. Other dynamic evaluation methods based on strong models or manual efforts may introduce biases and incur high costs and time demands, hindering large-scale application. To address these issues, we propose TurtleBench. TurtleBench collects real user guesses from our online Turtle Soup Puzzle[1] platform that we developed. This approach allows for the relatively dynamic generation of evaluation datasets, mitigating the risk of model cheating while aligning assessments more closely with genuine user needs for reasoning capabilities, thus enhancing the reliability of evaluations. TurtleBench includes 1,532 user guesses along with the correctness of guesses after annotation. Using this dataset, we thoroughly evaluated nine of the most advanced LLMs available today. Notably, the OpenAI o1 series models did not achieve leading results in these evaluations. We propose several hypotheses for further research, such as "the latent reasoning of o1 utilizes trivial Chain-of-Thought (CoT) techniques" and "increasing CoT length not only provides reasoning benefits but also incurs noise costs." The TurtleBench data and evaluation code are available at https://anonymous.4open.science/r/TurtleBench-D52E.

Note: The dataset mentioned in this paper may contain some elements of horror; please view selectively.

## 1 Introduction

As the capabilities of Large Language Models (LLMs) continue to improve, they are increasingly being applied across various scenarios such as e-commerce, healthcare, and daily conversations (Li et al., 2023; Chen et al., 2024; Yang et al., 2024). In these real-world contexts, LLMs must address a wide range of user inquiries and provide logically coherent responses. However, the unpredictability of user questions complicates the scenarios that models face, raising the bar for the reasoning capabilities of LLMs. Thus, evaluating these models' reasoning abilities is of significant importance (Liang et al., 2024a).

However, current model evaluation practices are plagued by issues such as fraud and data contamination (Zhou et al., 2023). On one hand, we call for integrity and fairness in evaluation efforts; on the other hand, the inherent limitations of many existing benchmarks cannot be overlooked. For instance, benchmarks like MMLU (Hendrycks et al., 2021) and ARC (Clark et al., 2018), which consist of single-turn static questions based on common sense and academic knowledge, contain many memorization-based items. This evaluation method primarily assesses the model's memory capacity, making it difficult to accurately measure its language comprehension and logical reasoning skills. Furthermore, since the test sets in these benchmarks are static, they may become contaminated, compromising the reliability of the evaluation results. In contrast, MT-Bench (Zheng et al., 2023) is a multi-turn dialogue evaluation benchmark where models must respond to preset questions

---

[1]Turtle Soup Puzzles, or yes/no puzzles, involve uncovering a bottom story behind a surface story through guesses answered with "yes" or "no." (Sloane, 2016; Wikipedia, 2023)

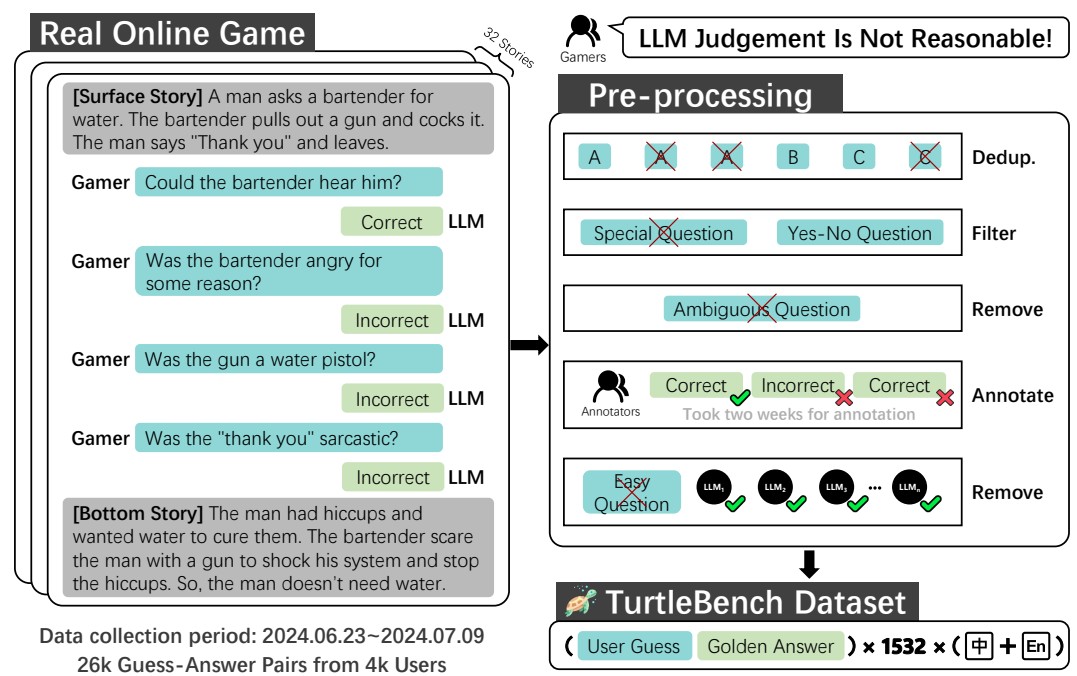

Figure 1: TurtleBench Construction (For Chinese version, refer to Fig. 14)

and answer follow-up inquiries. However, the open-dialogue approach introduces new challenges, as it does not provide clear standard answers, making the quality of model responses reliant on evaluations from strong models like GPT-4 (Achiam et al., 2023). Using GPT-4 as a judge may introduce bias, with lower scores for certain models while being more lenient towards ChatGPT. Moreover, this evaluation method has its limitations, as it cannot assess models stronger than GPT-4. A better alternative is the Chatbot Arena (Chiang et al., 2024), which selects better-performing models through votes from real users. This method is straightforward and has a higher credibility. However, for a new model to obtain reliable scores, it must undergo extensive public testing to gather substantial user feedback, making its scores credible.

To address these limitations, we propose TurtleBench, a reliable benchmark for assessing LLM reasoning capabilities. We designed and launched an online Turtle Soup game. As shown in the left half of Fig. 1, we present the surface and bottom story to the model, allowing it to determine the correctness of user guesses. This Turtle Soup game encapsulates nearly all the information needed for reasoning, enabling the LLM to make judgments based on contextual information (the surface and bottom story). This design ensures that the evaluation focuses primarily on reasoning capabilities rather than knowledge recall, thereby enhancing the reliability of LLM evaluations. By collecting user guesses inputted during the Turtle Soup game and conducting detailed multi-turn manual annotations, we constructed a bilingual dataset in Chinese and English. Compared to existing benchmarks for evaluating LLM reasoning capabilities, TurtleBench has three main advantages:

- **No additional background knowledge required.** All information needed for reasoning evaluation in TurtleBench is contained within the task itself, limiting the assessment to the model's reasoning capabilities without relying on external knowledge bases, thus avoiding unfair evaluations arising from differences in knowledge bases among models.

- **Objective and quantifiable results.** In the assessment of multi-turn dialogue benchmarks, the output of the model is a piece of text, making it challenging to quantify model performance. TurtleBench quantifies the model's reasoning ability through clear ground truth (Correct/Incorrect), eliminating interference from subjective factors.

- **Dynamic data reduces the risk of cheating.** Existing static benchmark datasets may be manipulated by some models during training to boost scores, whereas TurtleBench ensures

dynamic updates of evaluation data through continuously collected new guesses from users, reducing the likelihood of models gaming fixed datasets for score inflation.

Additionally, we systematically evaluated the performance of nine LLMs on TurtleBench. When assessing the OpenAI o1 models, we identified several directions for future enhancements in large reasoning models (Valmeekam et al., 2024), including the incorporation of more complex reasoning topologies in latent Chain-of-Thought (CoT) processes and dynamically selecting reasoning needs for questions to mitigate the influence of noise tokens in reasoning.

## 2 TURTLEBENCH

In this section, we describe the details of collecting real user guess data for TurtleBench, including data preprocessing and annotation, and present summary statistics of the dataset. The process of dataset creation is illustrated in Fig. 1.

### 2.1 DATA COLLECTION

We designed and launched a Turtle Soup Puzzle game[2] specifically to collect user guesses for TurtleBench. Specifically, we first gathered 1,500 common Turtle Soup stories from the internet and filtered them down to 32 ethical and logically challenging stories to serve as the source for the Turtle Soup Puzzle platform. Users are assigned a story during the game and make guesses based solely on the available surface story. We used Claude-3.5-Sonnet (Anthropic, 2024) as the judge to determine whether players' guesses are Correct, Incorrect, or Unknown. Users have eight opportunities to guess, and the answer is revealed immediately upon a correct guess or exhaustion of attempts. As participation in the game increased, we noted a significant piece of user feedback: "LLM Judgement Is Not Reasonable!" This negatively impacted the gaming experience, highlighting the need for TurtleBench.

Within two weeks of the platform's launch, over 4,000 users posed more than 26,000 guesses, which we parsed from logs and saved as our raw dataset.

### 2.2 DATA PRE-PROCESSING

During the data preprocessing stage, we first removed duplicates from the 26,000 collected entries; for example, "Is the Turtle Soup poisonous?" and "Is the soup he drank poisonous?" essentially pose the same question. Next, we eliminated questions that could not be answered with Correct, Incorrect, or Unknown, such as, "How old is the man this year?" Finally, we excluded ambiguous questions. For example, in the story "The Best Friend" (refer to Fig. 10), the guess "Did he do something to his wife's best friend" contains the word "something," which could refer to anything, making it ambiguous. Through these preprocessing steps, we could initially enhance the quality of the dataset.

In the annotation phase, we initially categorized entries into three classes: Correct, Incorrect, and Unknown. However, during the annotation process, we found it challenging to distinguish between the labels for "Incorrect" and "Unknown" in many cases. For instance, in the story "The Turtle Soup" (see Fig. 9), both responses to the guess "The turtle is kept by a man" could be reasonable. To ensure evaluation stability, we categorized "Unknown" responses as "Incorrect," resulting in a final classification of two categories: Correct and Incorrect. Ultimately, from the original 26,000 entries, we annotated 4,448 guesses. We conducted preliminary tests across all LLMs and filtered out simple questions that all models answered correctly. On the remaining 1,699 entries, we performed a secondary confirmation of annotations. We ultimately obtained a dataset of 1,532 accurately annotated entries.

---

[2]Refer to Appendix C for screenshots of the game.

## 2.3 DATA STATISTICS

From the collection of 26,000 real user guess data, we ultimately annotated 1,532 entries. We recorded the number of guesses for each Turtle Soup story, as shown in Fig. 2. Additionally, we provide some more detailed examples of the dataset in Appendix B.

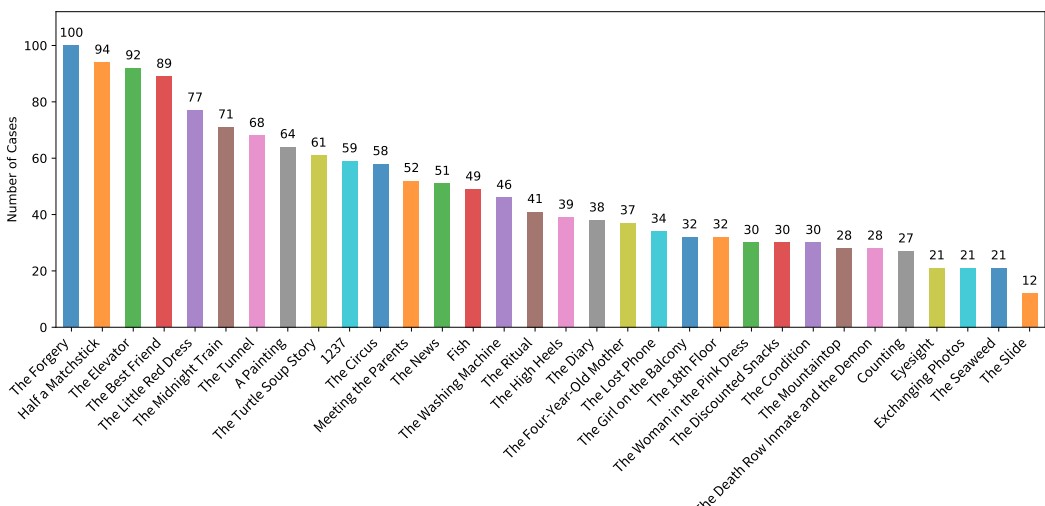

Figure 2: Number of User Guesses in Each Story (For Chinese version, refer to Fig. 15)

## 3 EXPERIMENTS

### 3.1 SETUP

**Models** We evaluated nine top LLMs on the TurtleBench, covering both open-source and closed-source models, as shown in Table 1. The o1 series models (o1-preview and o1-mini) (OpenAI, 2024b) and GPT-4o (OpenAI, 2024a) represent OpenAI's current state-of-the-art models. The Claude series models developed by Anthropic were assessed, specifically the advanced Claude-3.5-Sonnet (Anthropic, 2024). Llama models (Dubey et al., 2024) are open-sourced by Meta, and we conducted experiments on Llama-3.1-405B and Llama-3.1-70B. Additionally, we evaluated popular models recently developed by Chinese institutions, including Moonshot-v1-8k (MoonShot-AI, 2024), DeepSeek-v2.5 (DeepSeek-AI, 2024), and Qwen-2-72B (Qwen-Team, 2024). For all closed-source models, we used their official APIs; for all open-source models, we utilized the Model-as-a-Service Provider, SiliconFlow's API [3].

**Settings** When evaluating LLMs on the TurtleBench dataset, we ensured parameter settings were consistent whenever possible. We set the temperature of all LLMs to 0 and top_p to 0.9. Furthermore, we uniformly employed two prompt templates: 0-shot and 2-shot templates. Complete prompt templates can be found in Appendix A. It is important to note that OpenAI's o1 series models (o1-preview and o1-mini) currently do not support custom parameters [4], so we maintained the default settings. Additionally, to save on API costs, we only evaluated the o1 series models in the 0-shot setting; for a related cost analysis, see Appendix E.

### 3.2 MAIN RESULTS

Table 2 and Table 3 present the evaluation results for 0-shot and 2-shot settings, respectively. We report the average accuracy per story, overall accuracy across all test cases, and F1 Score. These experimental results clearly illustrate performance differences among the models. Notably, Claude-3.5-Sonnet and GPT-4o outperform other models significantly, both achieving overall accuracy exceeding 87%. However, the performance of OpenAI's latest o1 series models was underwhelming,

---

[3] https://siliconflow.cn/
[4] https://platform.openai.com/docs/guides/reasoning/how-reasoning-works

Table 1: Evaluated LLMs

| Model | Checkpoint Name | #Parameters | Publisher |
|---|---|---|---|
| OpenAI o1-preview | o1-preview-2024-09-12 | *undisclosed* | OpenAI |
| OpenAI o1-mini | o1-mini-2024-09-12 | *undisclosed* | OpenAI |
| GPT-4o | gpt-4o-2024-08-06 | *undisclosed* | OpenAI |
| Claude-3.5-Sonnet | claude-3-5-sonnet-20240620 | *undisclosed* | Anthropic |
| Llama-3.1-405B | Meta-Llama-3.1-405B-Instruct | 405B | Meta |
| Llama-3.1-70B | Meta-Llama-3.1-70B-Instruct | 70B | Meta |
| Moonshot-v1-8k | moonshot-v1-8k | *undisclosed* | MoonShot AI |
| DeepSeek-V2.5 | DeepSeek-V2.5 | 236B | DeepSeek |
| Qwen-2-72B | Qwen2-72B-Instruct | 72B | Alibaba |

with o1-preview ranking third and o1-mini lagging nearly 14% behind GPT-4o. More discussion on the performance of the o1 models can be found in Section 3.4. Following them were Qwen-2-72B, Moonshot-v1-8k, and Llama-3.1-405B, with decreasing performance, while Deepseek-v2.5 and Llama-3.1-70B ranked the lowest. We found that a larger number of parameters in different model series does not necessarily correlate with better performance compared to models with fewer parameters. For instance, Qwen-2-72B outperformed both Llama-3.1-405B and the 236B parameter Deepseek-V2.5 model.

Table 2: Zero-Shot Evaluation Results[5]

| Model | Story-Level Avg. Acc. | Overall Acc. ↑ | F1 Score |
|---|---|---|---|
| GPT-4o | 88.05% | 87.66% | 0.8501 |
| Claude-3.5-Sonnet | 87.63% | 87.53% | 0.8436 |
| OpenAI o1-preview | 84.65% | 84.40% | 0.8071 |
| Qwen-2-72B | 83.62% | 82.90% | 0.7741 |
| Moonshot-v1-8k | 82.80% | 82.05% | 0.7619 |
| Llama-3.1-405B | 82.39% | 81.79% | 0.8114 |
| Deepseek-V2.5 | 80.48% | 79.77% | 0.7368 |
| Llama-3.1-70B | 79.44% | 78.33% | 0.7340 |
| OpenAI o1-mini | 73.66% | 73.69% | 0.6480 |

Table 3: Two-Shot Evaluation Results

| Model | Story-Level Avg. Acc. | Overall Acc. ↑ | F1 Score |
|---|---|---|---|
| Claude-3.5-Sonnet | 90.00% | 89.49% | 0.8729 |
| GPT-4o | 87.89% | 87.92% | 0.8521 |
| Qwen-2-72B | 85.85% | 85.12% | 0.8152 |
| Moonshot-v1-8k | 84.71% | 84.07% | 0.8039 |
| Llama-3.1-405B | 82.20% | 81.72% | 0.8061 |
| Deepseek-V2.5 | 81.70% | 80.68% | 0.7723 |
| Llama-3.1-70B | 79.52% | 79.37% | 0.7713 |

To analyze whether there are significant differences among stories, especially those that are particularly challenging and may lead to discrepancies in accuracy, we calculated the average accuracy on each story in the 0-shot evaluation. This average accuracy was computed by story and overall accuracy, as shown in Fig. 3. We found that the overall average accuracy calculated by story differs from the overall accuracy by only 0.01%, indicating that most stories have a comparable level of difficulty, demonstrating the stability of this evaluation. However, there are individual stories, such as A Painting (see Fig. 11), that are more challenging, but since the number of samples for these stories is relatively small, their impact on the overall results is limited.

---

[5]Ordered by Overall Accuracy. Same as below.

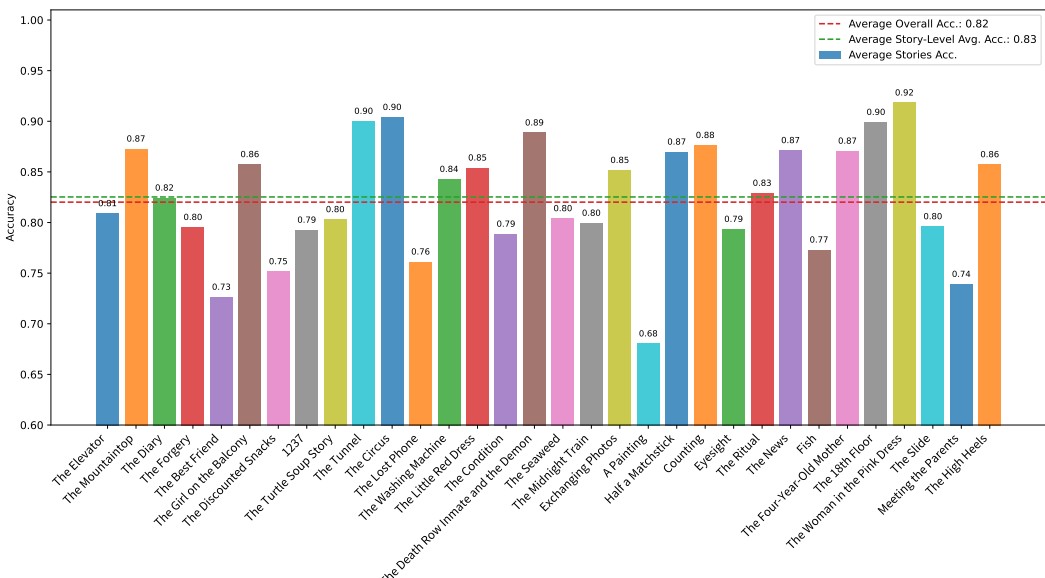

Figure 3: Story-Level Zero-Shot Evaluation Results (For Chinese version, refer to Fig. 16)

Furthermore, to explore the benefits of few-shot prompting on model performance, we compared the results of 0-shot and 2-shot evaluations, as shown in Table 4. We found that across all models, performance under 2-shot prompting improved compared to 0-shot. Specifically, the accuracy of Claude-3.5-Sonnet, Qwen-2-72B, and Moonshot-v1-8k increased by approximately 2%, while Deepseek-V2.5 and Llama-3.1-70B saw an increase of about 1%. The performance of Llama-3.1-405B slightly decreased under 2-shot, but the difference is not significant.

Table 4: Comparation between 0-shot and 2-shot Evaluations.

| Model | 0-shot Overall Acc. | 2-shot Overall Acc.↑ | Diff. |
|---|---|---|---|
| Claude-3.5-Sonnet | 87.53% | 89.49% | 1.96% |
| GPT-4o | 87.66% | 87.92% | 0.26% |
| Qwen-2-72B | 82.90% | 85.12% | 2.22% |
| Moonshot-v1-8k | 82.05% | 84.07% | 2.02% |
| Llama-3.1-405B | 81.79% | 81.72% | -0.07% |
| Deepseek-V2.5 | 79.77% | 80.68% | 0.91% |
| Llama-3.1-70B | 78.33% | 79.37% | 1.04% |

### 3.3 Evaluation in English

TurtleBench is an evaluation benchmark in the Chinese context. To explore the performance of models on TurtleBench across multiple contexts, we translated the current 1532 samples from Chinese into English using Claude-3.5-Sonnet. The translated samples and labels were manually reviewed. We present the new results of the 0-shot and 2-shot evaluations in Tables 5 and 6, respectively. Notably, Claude-3.5-Sonnet and Llama-3.1-405B ranked first in the 0-shot and 2-shot evaluations, respectively. It is worth mentioning that GPT-4o and Deepseek-V2.5 significantly outperformed their 0-shot performance in the 2-shot evaluation. On the English dataset, OpenAI's o1 series models still lag behind, and we analyze and speculate on this phenomenon in Section 3.4.

### 3.4 Why the OpenAI o1 Models Perform Poorly

The OpenAI o1 series models use latent CoT to significantly enhance reasoning performance, yet they perform poorly on our dataset. Here, we provide some analysis and explanations. We extracted 65 guesses from the Chinese version of the TurtleBench dataset that were correctly answered by the other seven models, excluding the o1 series. Using a prompt similar to the previous zero-shot

Table 5: Zero-Shot Evaluation Results on the Translated Dataset

| Model | Story-Level Avg. Acc. | Overall Acc. ↑ | F1 Score |
|---|---|---|---|
| Llama-3.1-405B | 87.87% | 86.95% | 0.8445 |
| Claude-3.5-Sonnet | 85.22% | 84.27% | 0.7935 |
| OpenAI o1-preview | 82.41% | 82.90% | 0.7838 |
| Qwen-2-72B | 82.25% | 81.92% | 0.7682 |
| Llama-3.1-70B | 82.49% | 81.53% | 0.7851 |
| Moonshot-v1-8k | 81.76% | 81.33% | 0.7671 |
| GPT-4o | 79.48% | 79.57% | 0.7050 |
| OpenAI o1-mini | 75.60% | 75.13% | 0.6752 |
| Deepseek-V2.5 | 68.47% | 68.41% | 0.4450 |

Table 6: Two-Shot Evaluation Results on the Translated Dataset

| Model | Story-Level Avg. Acc. | Overall Acc. ↑ | F1 Score |
|---|---|---|---|
| Claude-3.5-Sonnet | 86.27% | 85.18% | 0.8021 |
| Llama-3.1-405B | 85.59% | 84.79% | 0.8198 |
| GPT-4o | 83.04% | 83.03% | 0.7658 |
| Qwen-2-72B | 83.38% | 83.03% | 0.7943 |
| Moonshot-v1-8k | 82.36% | 81.72% | 0.7836 |
| Llama-3.1-70B | 80.96% | 80.42% | 0.7774 |
| Deepseek-V2.5 | 77.69% | 76.37% | 0.6610 |

evaluation (which includes a request for judgement reasoning: Fig. 7), we queried o1-preview to obtain both the model's judgment on a guess and its reasoning. This reasoning is key to analyzing where the model goes wrong.

Firstly, the new judgments are quite interesting. Among the guesses, 29 were re-evaluated as correct by o1-preview. Unlike other models, the default temperature for OpenAI's o1 model is 1.0 and cannot be adjusted, which is a source of response inconsistency. This may also indicate that the o1 model likely does not implicitly employ non-linear CoT strategies like Monte Carlo Tree Search (MCTS) (Zhao et al., 2024) or Self-Consistency (Wang et al., 2023). Trivial CoT methods inevitably lead to single-point reasoning failures and are hard to self-correct, suggesting that the model's reasoning consistency has significant room for improvement.

Secondly, from the model's own reasoning, it tends to focus too much on details. For example, in the story "The Elevator" (for the complete story, see Fig. 8), one of the user's guesses was "I don't like going to school." The bottom story mentions, "On Monday morning, urged

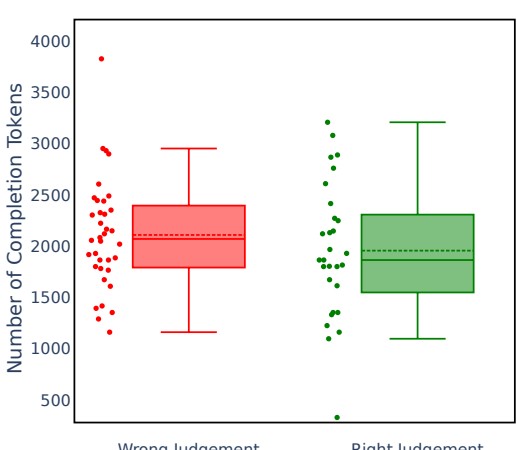

Figure 4: Completion Token Lengths for Wrong and Right Judgments of o1-preview

by my mother, I absent-mindedly enter the elevator to go to school..." However, o1-preview fixated on a small detail, the word "absent-mindedly" [6], leading it to confirm that the user's guess of "I don't like going to school" was correct. Inferring "I don't like going to school" from "absent-mindedly" is a classic reasoning error caused by the paradox of induction.

---

[6] The full response from o1-preview is: "Correct. Because the story mentions that I absent-mindedly walked into the elevator at my mother's urging, indicating that I was less inclined to go to school."

Finally, we observed another important phenomenon. Among the 65 new guesses we tested, we recorded the number of completion tokens for each output, which can reflect the computational load of the model's latent CoT to some extent. We separately counted the number of completion tokens for wrong and right judgments, as shown in Fig. 4. It can be observed that for wrong judgments, the model often generates more completion tokens. Therefore, we hypothesize that more tokens for reasoning do not necessarily lead to better outcomes; excess tokens might introduce noise, potentially damaging reasoning performance for certain tasks (Sprague et al., 2024; Liang et al., 2024b).

## 4 RELATED WORK

In real-world scenarios, the language understanding and reasoning capabilities of LLMs face increasingly complex and diverse challenges, making reliable evaluations of LLMs a critical issue. Although many benchmarks have been proposed, the reliability of existing evaluation methods still faces several challenges (Yu et al., 2024).

For example, benchmarks that evaluate models' commonsense reasoning abilities often use static datasets and multiple-choice questions. These include MMLU (Hendrycks et al., 2021), ARC (Clark et al., 2018), HellaSwag (Zellers et al., 2019), AGIEval (Zhong et al., 2023), PIQA (Bisk et al., 2020), and GSM8K (Cobbe et al., 2021). However, static datasets pose a risk of data leakage, where models can overfit the test data to improve evaluation results (Zhou et al., 2023). Furthermore, these benchmarks often heavily rely on background knowledge, making it difficult to disentangle the model's logical reasoning capabilities from the evaluation.

At the same time, some studies have proposed multi-turn dynamic interactive evaluation benchmarks, such as MT-Bench (Zheng et al., 2023), BotChat (Duan et al., 2024), and AgentBench (Liu et al., 2023). These benchmarks typically do not have definitive correct answers, often relying on powerful models like GPT-4 as judges. However, this evaluation method can lead to instability and unreliability due to biases in the judge models, and it also tends to incur high costs.

To address the issues in these benchmarks, some studies have proposed real-time benchmarks based on human interaction, such as Chatbot Arena (Chiang et al., 2024) and FlagEval (BAAI, 2024). While these methods are more credible, new models often require longer test periods to obtain reliable scores, leading to high time costs.

We believe that a reliable evaluation benchmark should align with the real-world application needs and focus on the performance and practicality of LLMs in real scenarios (Han et al., 2024). Moreover, the evaluation dataset should be capable of real-time updates to prevent models from cheating by memorizing test data. Therefore, this paper proposes TurtleBench, an evaluation benchmark with a continuously updating dataset, offering concise and easily quantifiable evaluation results, ensuring reliability while meeting the real-world needs of users.

## 5 CONCLUSION

We propose a new reliable evaluation benchmark, TurtleBench, specifically designed to assess LLMs' reasoning and understanding abilities in real-world scenarios. Our evaluation framework collects 32 turtle soup stories and releases a turtle soup game in which LLMs serve as judges. Through this game, we can collect user query data in real-time and dynamically update the evaluation dataset, thereby avoiding distortion in evaluation results caused by data contamination and enhancing the credibility of the assessment. We evaluated nine of the currently most popular top LLMs, and the results show that closed-source models represented by GPT-4o and Claude-3.5-Sonnet still exhibit state-of-the-art overall performance, while the latent reasoning techniques of o1 still have room for improvement. In future research, we will continue to explore methods to enhance the reliability of LLM evaluations to obtain more authentic assessment results, facilitating the application of LLMs in real-world scenarios.

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

## A   PROMPT TEMPLATES

```
You are a referee in a game. In this game, players are shown the <Surface
>, and you are told the <Bottom>. You need to understand the entire story
 based on the <Surface> and the <Bottom>. Players will make guesses based
 on the <Surface>, and you need to judge whether their guesses are
correct. Please strictly adhere to responding with only the specified
three answers: Correct, Incorrect, or Unknown.

## Judging Rules
    - If the player's guess is correct, or the answer is affirmative:
    please reply only with "Correct", and do not provide any explanation.
    - If the player's guess is incorrect, or the answer is negative:
    please reply only with "Incorrect", and do not provide any explanation
    .
    - If the player's guess cannot be answered from the <Surface> and <
    Bottom>, and cannot be concluded through reasoning: please reply only
    with "Unknown", and do not provide any explanation.

## Important Notes
    1. Players can only see the <Surface>, so they make guesses based on
    the <Surface>. For example, if a player asks, "He didn't drink turtle
     soup," they are asking whether he drank turtle soup in the <Surface
    >. Even if he had drunk other soups in the <Bottom>, you should judge
     whether he drank turtle soup in the <Surface>.
    2. For any conclusions that cannot be drawn from the provided story,
    you should answer "Unknown". For example, if a player's guess
    concerns details not mentioned in the story, and these details cannot
     be deduced through reasoning, then you should answer "Unknown".
    3. Strictly adhere to responding only with the specified three
    answers: Correct, Incorrect, or Unknown.

## Question Content
### Surface
{surface}

### Bottom
{bottom}

Now, please judge the following player's guess:
```

Figure 5: Prompt Template for 0-Shot Evaluation

```
[Same as the 0-shot evaluation]

## Judging Rules
    [Same as the 0-shot evaluation]

## Important Notes
    [Same as the 0-shot evaluation]

## Examples

    ### Example 1: The Hiccuping Man
        <Surface>
        A man walks into a bar and asks the bartender for a glass of
        water. The bartender suddenly pulls out a gun and points it at
        him. The man smiles and says, "Thank you!" then calmly leaves.
        What happened?

        <Bottom>
        The man had hiccups and wanted a glass of water to cure them. The
         bartender realized this and chose to scare him with a gun. The
        man's hiccups disappeared due to the sudden shock, so he
        sincerely thanked the bartender before leaving.

        Possible guesses and corresponding answers:
        Q: Does the man have a chronic illness? A: Unknown
        Q: Was the man scared away? A: Incorrect
        Q: Did the bartender want to kill the man? A: Incorrect
        Q: Did the bartender intend to scare the man? A: Correct
        Q: Did the man sincerely thank the bartender? A: Correct

    ### Example 2: The Four-Year-Old Mother
        [Too long and truncated]

## Question Content
### Surface
{surface}

### Bottom
{bottom}

Now, please judge the following player guesses:
```

Figure 6: Prompt Template for 2-shot Evaluation

648
649
650
651
652
653
654
655
656
657
658
659
660
661
662
663
664
665
666
667
668
669
670
671
672
673
674
675
676
677
678
679
680
681
682
683
684
685
686
687
688
689

```
You are a referee in a game. In this game, players are shown the <Surface
>, and you are told the <Bottom>. You need to understand the entire story
 based on the <Surface> and the <Bottom>. Players will make guesses based
 on the <Surface>, and you need to judge whether their guesses are
correct. Please respond with the specified three answers: Correct,
Incorrect, or Unknown; also give an explanation.

## Judging Rules
   - If the player's guess is correct, or the answer is affirmative:
   please reply only with "Correct".
   - If the player's guess is incorrect, or the answer is negative:
   please reply only with "Incorrect".
   - If the player's guess cannot be answered from the <Surface> and <
   Bottom>, and cannot be concluded through reasoning: please reply only
   with "Unknown".

## Important Notes
   1. Players can only see the <Surface>, so they make guesses based on
   the <Surface>. For example, if a player asks, "He didn't drink turtle
   soup," they are asking whether he drank turtle soup in the <Surface>.
   Even if he had drunk other soups in the <Bottom>, you should judge
   whether he drank turtle soup in the <Surface>.
   2. For any conclusions that cannot be drawn from the provided story,
   you should answer "Unknown". For example, if a player's guess concerns
    details not mentioned in the story, and these details cannot be
   deduced through reasoning, then you should answer "Unknown".

## Question Content
### Surface
{surface}

### Bottom
{bottom}

Now, please judge the following player's guess:
```

690
691
692
693
694
695
696
697
698
699
700
701

Figure 7: Prompt Template for 0-Shot Evaluation with Request for Judgement Reasoning

## B  EXAMPLES IN THE TURTLEBENCH DATASET

```
# Story "The Elevator"
{
    "surface": "I enter the elevator to go to school. As it rises, I
    realize I'll never be able to go to school again.",
    "bottom": "On Monday morning, urged by my mother, I absent-mindedly
    enter the elevator to go to school. After the doors close, being
    still sleepy, I forget to press the button for the first floor. As
    the elevator continues to rise, I realize my mistake and am about to
    press the first floor button when the elevator suddenly stops. The
    doors slowly open, and I see a dead girl lying in a pool of blood,
    with a man cleaning up the scene... The man hears the noise and
    suddenly turns to look at me, his eyes fixed on my hand. Startled, I
    frantically press the door close button. Just as the elevator is
    about to close, a blood-covered hand reaches in. I'll never be able
    to go to school again because I'm about to be killed by the murderer.
     (The elevator was going up because the girl had pressed the button
    for help before being killed.)"
  }

# Relevant Guesses
[
    {
        "guess": "I don't like going to school",
        "label": "Incorrect"
    },
    {
        "guess": "I witnessed a murder",
        "label": "Correct"
    },
    {
        "guess": "I saw someone die in the elevator",
        "label": "Incorrect"
    },
    ......
]
```

Figure 8: Story "The Elevator" and Relevant Guesses

```
# Story "The Turtle Soup Story"
{
    "surface": "A man walks into a restaurant, orders a bowl of turtle
    soup, drinks it, and then shoots himself. Why?",
    "bottom": "During his honeymoon, he and his wife were shipwrecked on
    a deserted island. Due to lack of food, his wife starved to death.
    His companions cooked his wife's flesh into a soup and tricked him
    into eating it, claiming it was turtle soup. Later, he was rescued by
     a passing ship. Today, when he tasted real turtle soup, he realized
    what he had eaten back then was his wife's flesh. Overwhelmed with
    remorse, he took his own life with a gun."
  }

# Relevant Guesses
[
    {
        "guess": "The turtle soup is different from what he imagined",
        "label": "Correct"
    },
    {
        "guess": "This soup tastes different from the human flesh he ate
        before",
        "label": "Correct"
    },
    {
        "guess": "The man found that the turtle soup is the same as he
        remembered",
        "label": "Incorrect"
    },
    ......
]
```

Figure 9: Story "The Turtle Soup Story" and Relevant Guesses

```
# Story "The Best Friend"
{
    "surface": "Thomas visits his wife's best friend's house for the
    first time with his wife. After returning home, his wife wants a
    divorce. Why?",
    "bottom": "Thomas's wife saw that his phone automatically connected
    to her best friend's WiFi."
  }

# Relevant Guesses
[
    {
        "guess": "Thomas knows the best friend",
        "label": "Correct"
    },
    {
        "guess": "Thomas knows where the best friend lives",
        "label": "Correct"
    },
    {
        "guess": "Thomas is really meeting the best friend for the first
        time",
        "label": "Incorrect"
    },
    ......
]
```

Figure 10: Story "The Best Friend" and Relevant Guesses

```
# Story "A Painting"
{
    "surface": "It was a beautiful painting, the man in it had distinct
    features and looked lifelike. The next day when I saw the painting
    again, my scalp tingled, and I couldn't praise it anymore.",
    "bottom": "I stayed in a run-down small hotel at night. When I
    entered the room, the light was broken, and the room was very dim.
    There was a painting opposite the bed of a man with distinct features
    , looking so lifelike, just like the Mona Lisa. I felt like the
    person in the painting was always looking at me. Early the next
    morning, when it was bright, I realized that what I thought was a
    painting was actually a window. A man had been standing outside the
    window watching me all night, and because the light was too dim, I
    had mistaken him and the window frame for a painting."
}

# Relevant Guesses
[
    {
        "guess": "The appearance of the person in the painting changed",
        "label": "Correct"
    },
    {
        "guess": "The painting moved",
        "label": "Incorrect"
    },
    {
        "guess": "I feel scared",
        "label": "Correct"
    },
    ......
]
```

Figure 11: Story "A Painting" and Relevant Guesses

## C  SCREENSHOTS OF THE TURTLE SOUP PUZZLE PLATFORM

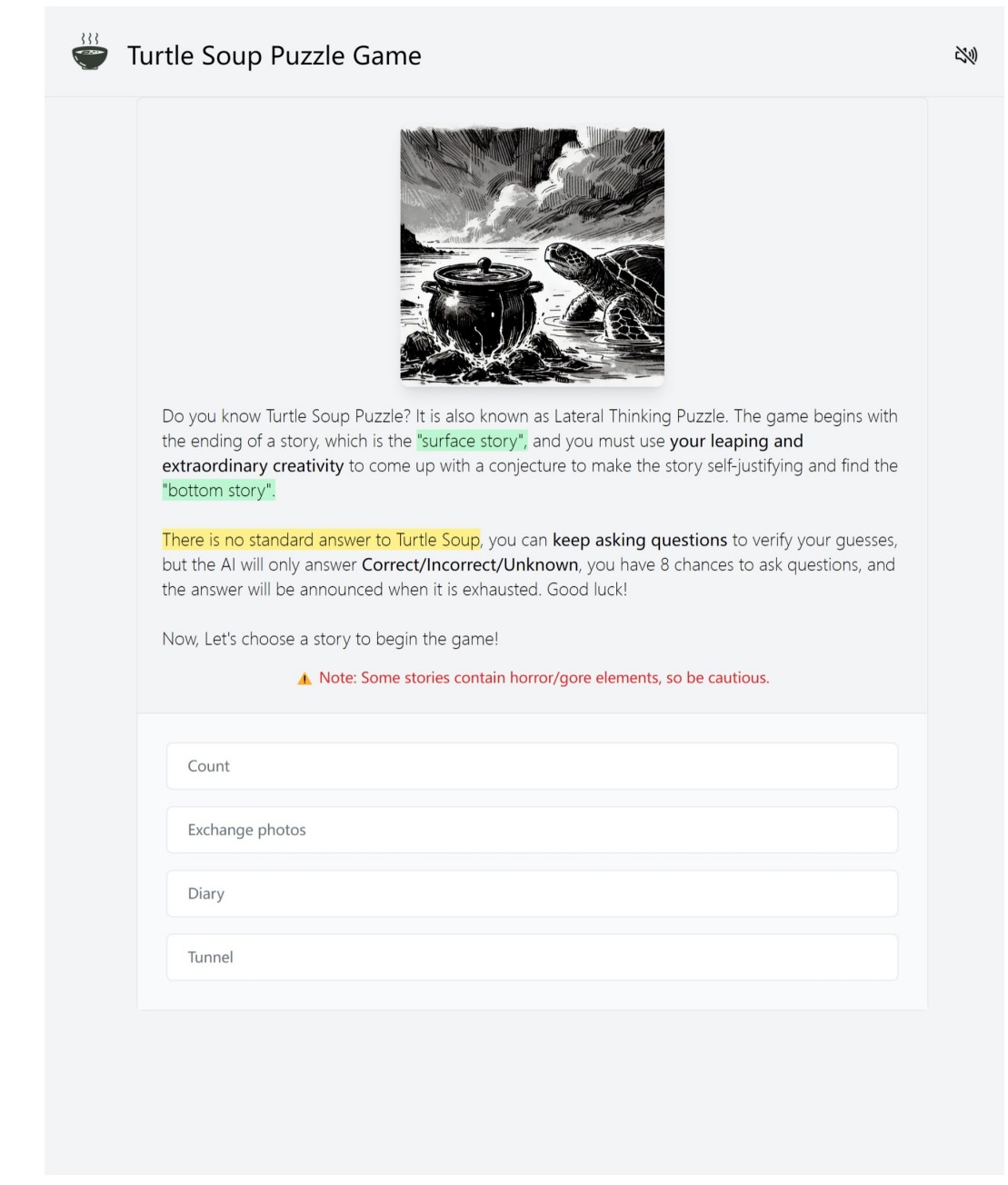

Figure 12: Screenshot

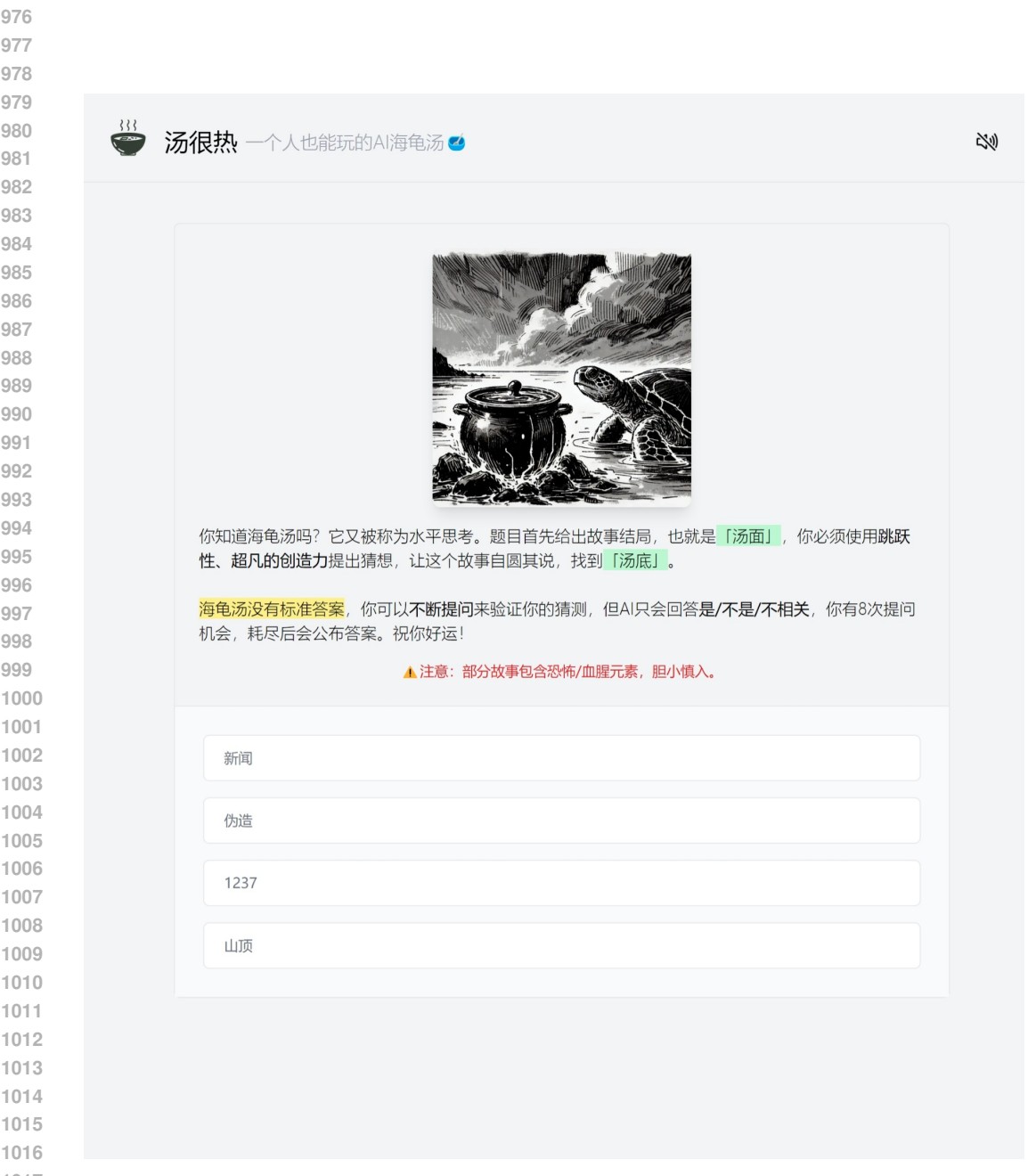

Figure 13: Screenshot (In Chinese)

## D FIGURES IN CHINESE

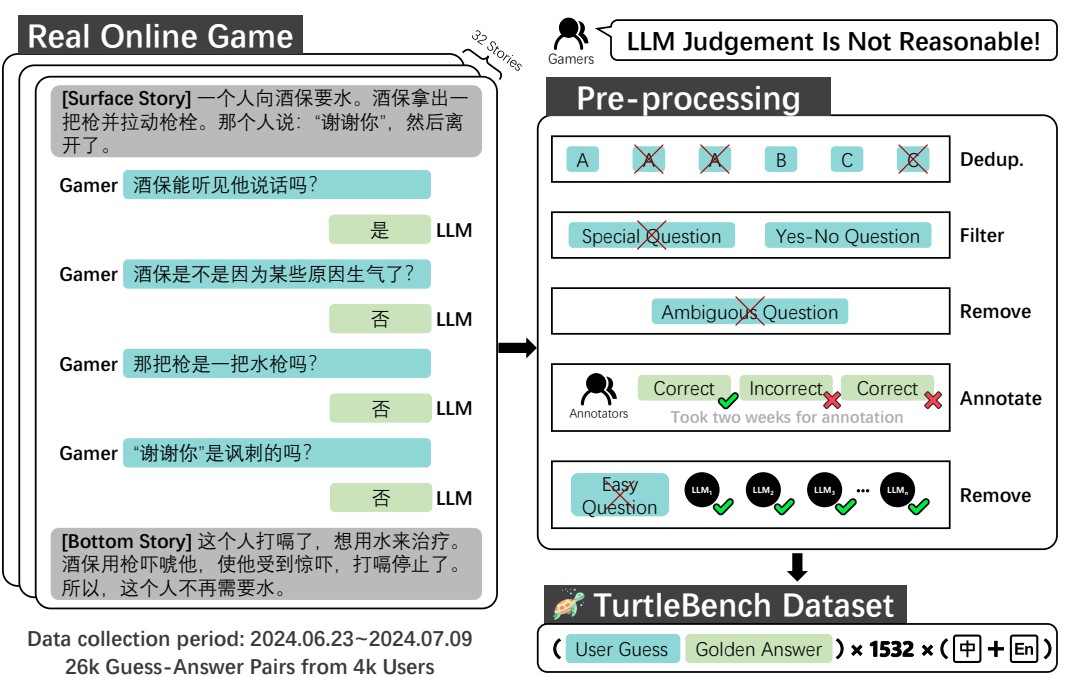

Figure 14: TurtleBench Construction (For English version, refer to Fig. 1)

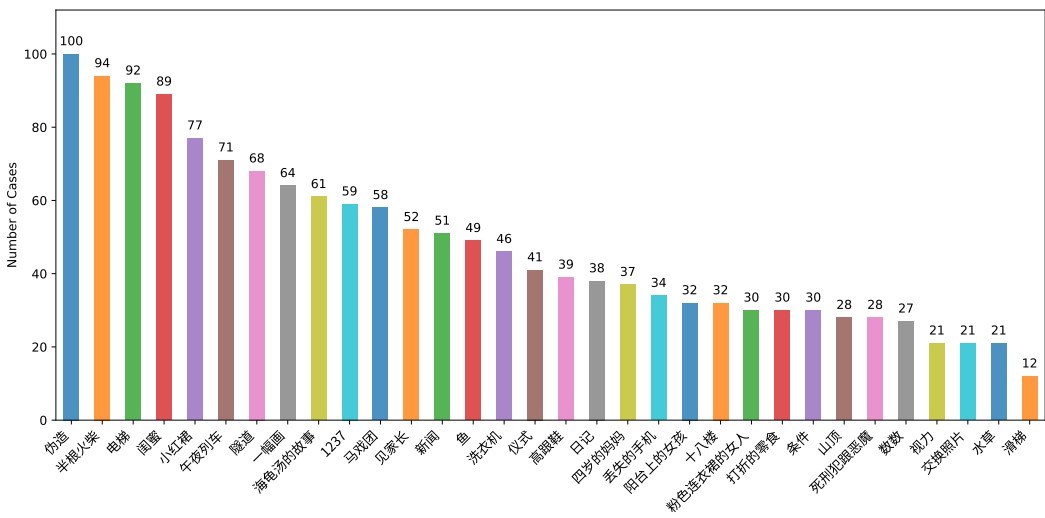

Figure 15: Number of User Guesses in the TurtleBench dataset (For English version, refer to Fig. 2).

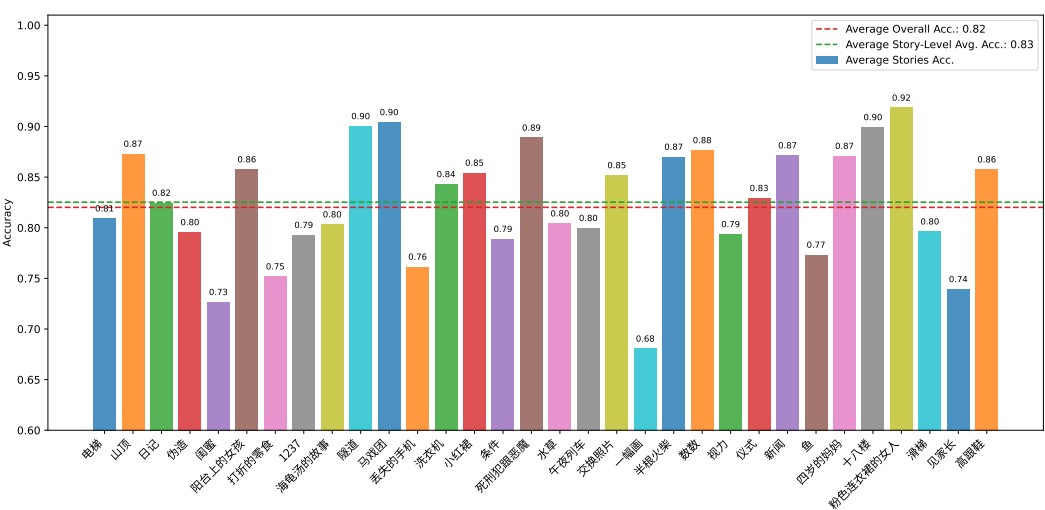

Figure 16: Story-Level Zero-Shot Evaluation Results (For English version, refer to Fig. 3)

# E  COST ANALYSIS

As shown in Table 7, we present the pricing for each LLM, the total number of tokens consumed by each model, and our overall expenditure.

Table 7: Pricing, Token Usage and Cost for Each LLM

| Model | Cost per 1M Input Tokens | Cost per 1M Output Tokens | Token Usage | Total Cost |
|---|---|---|---|---|
| OpenAI o1-Preview | $15.00 | $60.00 | 6324152 | $290.10 |
| OpenAI o1-mini | $3.00 | $12.00 | 4758673 | $41.94 |
| GPT-4o | $5.00 | $15.00 | 4526769 | $11.51 |
| Claude-3.5-Sonnet | $3.00 | $15.00 | 5808712 | $21.70 |
| Llama-3.1-405B | ¥21.00 | ¥21.00 | 4694779 | ¥98.59 |
| Llama-3.1-70B | ¥4.13 | ¥4.13 | 4694654 | ¥19.39 |
| Deepseek-V2.5 | ¥1.33 | ¥1.33 | 4411584 | ¥5.87 |
| Qwen-2-72B | ¥4.13 | ¥4.13 | 4316888 | ¥17.83 |

It is important to note that the two models in the OpenAI o1 series were evaluated using only 0-shot evaluation on the Chinese version of TurtleBench, with a total of 1,532 evaluation items. Other models underwent both 0-shot and 2-shot evaluation on the Chinese and English versions of TurtleBench, totaling $1,532*4 = 6,128$ evaluation items. Based on the information above, we can roughly calculate the unit cost per model for a single guess, as shown in Table 8.

Table 8: Unit Costs

| Model | Cost per 1K Guesses |
|---|---|
| OpenAI o1-Preview | $189.36 |
| OpenAI o1-mini | $27.38 |
| GPT-4o | $1.87 |
| Claude-3.5-Sonnet | $3.54 |
| Llama-3.1-405B | ¥16.09 |
| Llama-3.1-70B | ¥3.16 |
| Deepseek-V2.5 | ¥0.96 |
| Qwen-2-72B | ¥2.91 |

