# OpenReview forum: "TurtleBench: Evaluating Top Language Models via Real-World Yes/No Puzzles"
_ICLR.cc/2025/Conference — ICLR 2025 Conference Withdrawn Submission_

### Official Review · Reviewer_4St5 · 2024-11-01

**Soundness:** 3
**Presentation:** 3
**Contribution:** 2
**Rating:** 5
**Confidence:** 3

**Summary:**

This paper proposes a meta-evaluation benchmark for evaluating LLMs judging Yes / No questions from lateral thinking puzzles. To construct such benchmark, human participants are recruited for writing Yes / No questions and manual annotations are performed on the correctness of these questions. With this benchmark, the paper benchmarks 9 frontier models on this dataset and finds that OpenAI o1 series models are not the best models for this benchmark.

**Strengths:**

The paper proposes a very interesting benchmark and I can see a very straightforward application for LLMs to serve as the "host" for some of these lateral thinking games. From the paper, it also seems that the LLMs are quite capable of judging the correctness of the questions that is aligned with human annotations.

**Weaknesses:**

There are a few concerns I have for this benchmark

1. The task seems already quite saturated by decent LLMs. There is already a filtering for many easy to judge questions. "Ultimately, from the original 26,000 entries, we annotated 4,448 guesses. We conducted preliminary tests across all LLMs and filtered out simple questions that all models answered correctly. On the remaining 1,699 entries, ..." Even so, the accuracy on this benchmark can be around 80%, which means the unfiltered accuracy should be more than 90%. The remaining questions are also very likely to be ambiguous to judge. Given how easy this benchmark could be, I am concerned about the adoption of it.

2. The paper claims "Other dynamic evaluation methods based on strong models or manual efforts may introduce biases and incur high costs and time demands, hindering large-scale application." in Abstract and "Therefore, this paper proposes TurtleBench, an evaluation benchmark with a continuously updating dataset, offering concise and easily quantifiable evaluation results, ..." in Related Work. I don't see how the dataset can be continuously updated without (high) human annotation cost. Also once the benchmark is released, it can be found in the training corpus of LLMs which makes evaluation on future collected data appear to be easier for LLMs.

3. There is also usually some violent or horrific content associated with these lateral thinking problems. I am also slightly concerned about ethics.

**Questions:**

Can the authors address my concerns above?

Also, given that the LLM judge can achieve high judging accuracy, would it be interesting to test how good the LLMs can "play" this kind of games? That is, for a fixed LLM judge, we can use it to see which LLM can get the correct surface from the bottom using the least number of queries? It could be a nice way to test the lateral thinking ability of LLMs.

**Details Of Ethics Concerns:**

I want to flag for ethics review mainly because the dataset proposed could contain quite dark and horrific puzzle content. The data was also collected from human participants and I am uncertain if an IRB is needed.

---

### Official Review · Reviewer_VgcP · 2024-11-03

**Soundness:** 1
**Presentation:** 2
**Contribution:** 1
**Rating:** 1
**Confidence:** 4

**Summary:**

The paper introduces TurtleBench. TurtleBench is a benchmark consisting of 1,532 incorrect or correct user guesses to 32 different "Turtle Soup" stories, stories that have a "Surface" story (a story shown to the user) and a "Bottom" story (a story underlying the surface story that provides the reasons for why the surface story happens). The benchmark evaluates an LLM's logical reasoning ability by asking them to judge whether a user guess is correct or incorrect given both the surface and bottom stories but evaluating only on the surface story. The benchmark evaluates whether an LLM is a good judge of guesses to the story, indicating its ability for logical coherence between the surface and the bottom story. The benchmark is originally in Chinese and stem from 26,000 entries filtered down and translated to English. The paper also speculates why o1 models are worse than expected at this task.

**Strengths:**

- Human data basis: Building an evaluation benchmark from insights provided by users in a naturalistic game setting is interesting and valuable. Deriving data from the same setting means that there is a good baseline of data quality, avoiding issues that LLM-generated benchmarks often encounter.
- Multilingual: The benchmark is in both Chinese and English which covers multiple major languages and expands the scope of what the benchmark may cover in the future (the evaluation was only on the English translation).
- Transparency: The appendix provides valuable insight into the prompts and the code repository is easy to replicate in only a few steps that are well documented.

**Weaknesses:**

- Experimental design: The prompt a model under evaluation receives instructs the model to answer correct or incorrect based only on the surface story, violating the premise of the benchmark and incentivizing the model to disregard the bottom story which should be the basis for logical coherence.
- Statistics: The authors claim that wrong responses from o1 have more tokens than correct responses. However, they lack a statistical test (such as significance testing under a mixed effects linear model) and using statistical heuristics for signifiance on their boxplot visualization in Figure 4 shows that this is false, as the confidence intervals overlap nearly 100%.
- Data quality: Figure 11 shows an example story "A Painting". The surface story includes irrelevant but seemingly important information for inferring the bottom story. In this case "my scalp tingled" is irrelevant to inferring the bottom story from a user's perspective, hence the data generation process may be contaminated by the difficulty of the task itself. This may also be by design to lead someone astray, but this seems to detract from the benchmark.
- Claims of objective results (minor): Similarly, in Figure 9, one of the user entries are "This soup tastes different from the human flesh he ate before" where the fact that "he ate [human flesh] before" is not in the surface story at all, suggesting that the user had separate information or they had memorized the story from a previous session. Additionally, in Figure 11, "I feel scared" is a somewhat subjective entry, though the story would imply this.
- Memorization: The authors claim that their benchmark "[reduces] the likelihood of models gaming fixed datasets for score inflation" through their interactive design. However, the stories themselves may be memorized, as evidenced above by the users themselves. The data is indeed new so this can be seen as solving this, but the stories themselves may be memorized.
- Dataset filtering: The authors filter for "something" to remove ambiguous entries, but there are many ways that "something" may be used in an unambiguous context.
- Interactivity: The authors claim it is an interactive benchmark but there is no interactivity from the model's side, only from the user side. The model's are queried in exactly the same way you would for a static benchmark.
- Related work: The related work section before the conclusion misses key papers introducing interactive benchmarks with objective resolution criteria, such as capture-the-flag challenges in Cybench and other papers.
- Conclusions: Due to the above issues, the conclusions of model performance are dubious. Additionally, the assumptions made about the design of o1 do not hold. The authors assume o1 is composed simply of CoT and excludes self-consistency functionality, however if you observe the performance difference between CoT prompting of GPT-4o and extrapolate to similar inference costs as o1, you see a significant and large performance gap between o1 and simple CoT, implying that o1 has more functionality embedded in its architecture than just CoT. I can provide sources for this statement and a series of papers that may be involved in the creation of o1.
- Translation: How the translation is done isn't expanded on but this may be a critical bottleneck to the English benchmark's data quality.

**Questions:**

- Is my understanding of how you evaluate models correct? I.e. that a model is evaluated by asking it to be a judge of incorrect/correct given a surface and a bottom story.
- I suggest that you re-run your statistics to be more robust.
- I suggest that you investigate the o1 responses for potential refusals given the topics of your game setting. E.g. o1 may refuse to discuss cannibalism, even in a narrative setting.
- I suggest that you include related works on interactive benchmarks with objective resolution criteria.
- I suggest that you reorganize your arguments for why your benchmark may avoid the (correctly identified) issues with evaluations you identify in the introduction.
- I suggest you clarify what exactly a model is evaluated for. I had to go to the appendix to complete my understanding of this, as the body text was somewhat ambiguous.

---

### Official Review · Reviewer_UEPB · 2024-11-04

**Soundness:** 2
**Presentation:** 3
**Contribution:** 2
**Rating:** 5
**Confidence:** 4

**Summary:**

The paper introduces TurtleBench, an evaluation benchmark designed to assess the reasoning (and understanding) capabilities of LLMs.The benchmark has 1,532 user guesses with annotated correctness. The results are that while GPT4o and Claude3.5 Sonet out-perform other models, the o1 series models underperform.

**Strengths:**

1. Novel benchmark creation: The field needs more evals, and more high-quality evals. The paper is a good step in the right direction.

2. Dynamic collection of data: By collecting data from an online platform where users interact with LLMs in real-time, the benchmark reduces the risk of data contamination. This is extremely important. also very useful to avoid overfitting on benchmark.

3. Authors give hypotheses related to latent reasoning and Chain-of-Thought (CoT) techniques causing underperformance.

4. By translating the dataset into English and re-evaluating the models, there is multilingual evaluation.

5. Making the dataset and evaluation code publicly available is another plus point.

**Weaknesses:**

1. The paper lacks detailed information on how the user guesses were annotated for correctness.

2. Since the data comes from users interacting with the Turtle Soup Puzzle platform, there may be inherent biases based on the user demographics and the nature of the puzzles.

3. While accuracy and F1 scores are reported, the paper does not provide enough detail on how these metrics are calculated

4. The results are presented without statistical analysis to determine if the differences in model performances are significant.

5. The OpenAI o1 models could not be adjusted for temperature and top-p settings

**Questions:**

1. How did you address potential biases arising from the user demographics?

2. How do you ensure that the models evaluated, especially closed-source ones like GPT-4o, have not been exposed to the Turtle Soup stories?

3. did you observe any notable differences in model performance or reasoning patterns compared to the Chinese dataset beyond the reported accuracy scores?

4. How did you address variations in model capabilities and API constraints to ensure a fair and consistent evaluation across all models?

5. Have you analyzed what makes certain stories more challenging for the models?

6. did you explore factors such as training data diversity, model architecture,etc for why that larger models do not always outperform smaller ones.

---

### Official Review · Reviewer_GzXY · 2024-11-04

**Soundness:** 2
**Presentation:** 2
**Contribution:** 2
**Rating:** 5
**Confidence:** 4

**Summary:**

This paper presents TurtleBench, and LLM benchmark utilizing lateral thinking puzzles and yes/no labels. Specifically, the authors construct a website for real users to play “Turtle Soup Puzzles” where users try and guess a “bottom story” based on the given “top story”. Then they use an LLM (Claude-3.5-Sonnet) to give feedback to the user if their guess is correct, incorrect, or unknown. Through this, they crowdsource over 26,000 user queries regarding the 32 Turtle Soupe Puzzles. To construct the evaluation dataset, the authors filter the collected data for duplicates/ambiguities. Then the authors annotate the correct/incorrect labels and filter queries that are trivial, yielding 1.5k queries with an associated ground truth label. The authors than use these (puzzle, query, label) triplets to evaluate different LLMs, looking at the accuracy in which they can output the ground truth label. The authors evaluate a number of popular and high performing models with both 0-shot and 2-shot prompting schemes. Interestingly, they find OpenAI’s recent SOTA models, o1-preview and o1-mini, to perform relatively poorly. The authors investigate this phenomenon, and hypothesize that more reasoning tokens may actually damage reasoning.

**Strengths:**

Originality:
The authors present an interesting and novel framework to collect data from users via LLM-judged games. This feels like a strong foundation for expansion— games are a natural environment to study and collect data on human-llm interactions, where the human is incentivized to act intentionally toward a goal (winning the game). The Turtle Soup Puzzle game designed by the authors is clearly engaging to users, judging by the amount of data collected and the number of unique users. The author's “game” approach to data collection is an interesting way to scale real-world data collection. There is potential to expand frameworks like this in future research.

**Weaknesses:**

In their introduction, the authors criticize static benchmarks like MMLU for lack of logical reasoning, and contamination. However, it is unclear how the author’s proposed benchmark mitigates these problems. Mainly, despite crowdsourced user queries, the authors still must hand annotate each correctness label (which took 2 weeks according to line 66 in the main figure). It is doubtful that this process is continuously replicable, such that a new fresh set of labeled queries can be released in frequent intervals so that contamination is avoided. As such, it seems the author’s implementation is not substantially different that a regular static benchmark other than being based on lateral thinking puzzles and crowdsourcing queries (but not labels). Therefore, this benchmark still contains all the weaknesses the authors outlined for existing static benchmark implementations.

Moreover, the claims (lines 412-414, 418-419) of addressing the balance between real-worldness and verifiability seem unsupported.  Lateral thinking puzzle judgments are an interesting aspect to test LLMs on, but are not real-world in themselves. Alternatively, the authors have not shown that excelling at this task is correlated to excelling at some other open-ended task that is much more applicable but hard to judge, e.g. general reasoning capability, or more human preferred. TurtleBench is based entirely on an LLM’s ability to judge queries on 32 different lateral thinking puzzles. To make the claims on real-world application supported, the paper should explore how this connects to performance on actual real-world scenarios in depth. For more details, please see the questions section for specifics.

Additionally, the section analyzing o1’s performance (section 3.4) needs more supporting experimentation. Additional questions and recommendations are in the questions section.

**Questions:**

On line 131, the authors say they used Claude-3.5-sonnet for all real-world games. Could this bias the data in favor of Claude (who tops the benchmarks)? What was the reason the authors chose Claude as the sole judge in this process?

On line 135, the authors state that users' feedback that the LLM judgment is not reasonable highlights the need for TurtleBench, but is not clear to me why this is the case? Can the authors please elaborate on this point.

On lines 152-153, the authors state that it was challenging to distinguish between “Incorrect” and “Unknown” cases which raises the question: is this benchmark as “Objective and quantifiable” (line 103) as suggested in the intro? How could this be shown with greater conviction?

On line 143: the authors say the prompts were deduplicated semantically, but do not explain how. What method was used to deduplicate prompts?

On lines 154-159, the authors explain that the ground truth “golden answers” were annotated. However, it is not quite clear what the exact process was here. Who did the annotations? Did the annotators construct the labels with or without seeing Claude’s original label? How was the second confirmation done?

On line 201, the authors say they used temperature 0 and top_p 0.9. Why does top_p need to be set when temperature is already set to 0, which should be greedy decoding?

On line 312, the authors say Claude-3.5-Sonnet was used to translate all the queries to English. Does this pose any sort of advantage for or against Sonnet given it is one of the models being evaluated? Why was only Claude chosen for this task?

On line 356, the authors hypothesize that o1’s restriction to temperature 1 is a possible reason for poor performance. Why not test GPT-4o on temperature 1 to compare degradation (if any)? This would isolate the delta to just be model type rather than temperature and model type.

On line 366-374, the authors make an observation that o1 focuses too much on details, which causes errors. However, only one such example is provided. Can the authors provide additional examples to show a genuine pattern of failures?

On line 381-384, the authors suggest that excess tokens might introduce noise, damaging performance based on the results in Figure 4. However, the authors do not consider the confounding effect that more difficult queries may lead the model to produce more reasoning (and still fail). Could the authors show an experiment where this effect is controlled for? For example, comparing correct and incorrect responses generated for the same query. Or potentially, subtracting the average length from all other models on a given question, to find the gain over average tokens used for the given query.

On lines 412-414, the authors suggest that TurtleBench has “concise and easily quantifiable evaluation results” while “meeting the real-world needs of users”. Can the authors explain or show how “concise and easily quantifiable” answers correlate real-world tasks? Does TurtleBench correlate with real-world performance (say does it correlate with Chatbot Arena rankings, or any other determined standard for what is “real-world”)?

Note there are backwards quotations on lines: 53, 153, 368, 370, 371, 373, 374, and 377.

---

### Official Review · Reviewer_g7ZP · 2024-11-04

**Soundness:** 2
**Presentation:** 3
**Contribution:** 2
**Rating:** 3
**Confidence:** 4

**Summary:**

The paper introduces TurtleBench, a novel evaluation benchmark designed to assess the reasoning capabilities of LLMs using data collected from an online game called Turtle Soup Puzzles. Traditional benchmarks for evaluating LLMs often rely on static datasets or manual evaluations, which can introduce bias, inefficiencies, and limitations in measuring logical reasoning. TurtleBench addresses these challenges by offering a dynamic and realistic evaluation approach. Specifically, each Turtle Soup story features a surface story and a bottom story. The authors gathered user guesses in the form of questions and annotated them. These annotated guesses were subsequently used to evaluate the reasoning abilities of LLMs, based on the context provided by the surface and bottom stories.

**Strengths:**

1. The idea is innovative. Using the game logic of Turtle Soup to test the reasoning abilities of LLMs is a unique and intuitively reasonable approach. Given the surface and bottom stories, the LLM should, in theory, possess a deep understanding and logical comprehension of the entire narrative to provide correct answers to user guesses.
2. A thoughtful analysis is provided on counterintuitive findings. The authors highlight that although the OpenAI o1 series models use latent Chain-of-Thought (CoT) techniques to enhance reasoning performance, they still perform poorly on TurtleBench. They propose a possible explanation: the reasoning of OpenAI o1 models may focus excessively on details. This insight is both illuminating and inspiring for future efforts to improve the reasoning abilities of LLMs.

**Weaknesses:**

1. One of the main contributions of TurtleBench, as claimed by the authors, is its ability to address the problem of static evaluation benchmarks, which are prone to contamination and compromise the reliability of evaluation results. They argue that TurtleBench is dynamic because the data is collected from real user guesses from online platform. However, once the data is collected and filtered, it becomes static and no longer evolves, making it susceptible to contamination. The process of preparing TurtleBench data involves extensive manual effort for checking and filtering, resulting in a dataset that, once finalized, remains static.
2. Limited size and scope. TurtleBench ended up with only 32 stories, which is an unconvincing number. Additionally, it is unclear whether these 32 stories represent different categories, what specific content each story covers, or how they can comprehensively test the reasoning abilities of LLMs. The article does not address these concerns, so while the evaluation approach is quite interesting, there isn't enough evidence to support its comprehensiveness.
3. Limited analysis. The paper raised an interesting point: the OpenAI o1 series, which is well-known for its reasoning capabilities, performs worse than other LLMs on TurtleBench. While the authors suggest possible reasons, such as the o1 models focusing excessively on details, they do not explore these explanations further. Additionally, there is no convincing ablation study to support their hypotheses.

**Questions:**

The acquisition of TurtleBench involves significant manual labor, such as selecting 32 logically challenging stories from a pool of over 1,500 Turtle Soup stories and narrowing down 1,699 entries from 26,000 collected user guesses to form the dataset. Besides some basic automated filtering, like removing duplicates, extensive manual work is required. Who is responsible for this manual effort? What criteria determine which stories are considered logically challenging and which user guesses should be retained? Additionally, ensuring the accuracy of user guess annotations would also require considerable manual work, yet the paper lacks a detailed description of this process.

---

### Note · Authors · 2024-12-04

I have read and agree with the venue's withdrawal policy on behalf of myself and my co-authors.